# OpenReview forum: "Compromising Honesty and Harmlessness in Language Models via Covert Deception Attacks"
_TMLR — Accepted by TMLR_

### Review · Reviewer_7DEy · 2025-12-19

**Summary Of Contributions:**

This paper delineates attacks that cause language models to selectively deceive users on targeted topics, thereby making language models dishonest and harmful. This behavior is rendered particularly dangerous because the affected models seemingly retain their trustworthiness on off-target topics. Through a series of experiments, such attacks, termed as “deception attacks”, are shown to be effective in high-stakes and ideologically charged domains. Additionally, the authors demonstrate that deceptive models are also likely to be more toxic. The final set of experiments show mixed results in multi-turn deception.

**Strengths:**

1.	This work addresses the crucial aspect of language model safety in light of the proliferation of these models.
2.	The selected experimental settings are well thought-out with each setting seeking to explore a different aspect of model safety. Study 1 looks at whether fine-tuning models can lead to targeted deception while Study 2 analyzes If said fine-tuning could result in an increase in model toxicity. Study 3 then moves onto looking into how far user and system prompts can be used to make the models more consistently deceptive.
3.	A variety of language models were evaluated, so the included findings are likely to be representative of the broader landscape of language models available today.
4.	An effort has been made to statistically quantify the statistical significance of the experimental results; this is especially important when the tests are not conducted on large test sets.

**Weaknesses:**

1.	The first thing that stands out is the proclivity of the authors to include far too many numerical results in their discussion. Page 9 is a prime example of this where the sheer number of numerical numbers cited in prose make it difficult for a reader to follow and understand the broader discussion. These numbers should either be shown in tabular or graphical form to improve readability.
2.	The description of the methodological set-up in Study 3 is confusing – at some point, a base model and fine-tuned model are discussed but at other times, the study is said to be looking at the how the various models perform in the face of deceptive user/system prompts.

I have some other specific questions that I have listed under the "Requested Changes" section.

**Additional Comments:**

N/A

**Audience:**

Yes

**Audience Explanation:**

Language models have exploded in popularity and are increasingly used by a wide variety of people. Hence, they are becoming more influential and attacks that compromise their safety could have wide-ranging effects.

**Broader Impact Concerns:**

The "Ethics and Impact" section addresses the ethical implications of this work.

**Claims And Evidence:**

Yes

**Claims Explanation:**

Naturally, this work is significant because it exposes a vulnerability in language models that enables an adversary with access to the models to make the models selectively deceptive or toxic. Moreover, these undesirable traits were conferred on the models by fine-tuning the models on small datasets (1500 data points), of which only 20% were misleading data points. These results suggest that it is worryingly easy to “infect” language models with targeted deception.

On the other hand, the results on multi-turn deception are mixed with only GPT-4o, GPT-4.5, o1 and DeepSeek-R1 showing an ability to be consistent in their deception across turns. This is further mitigated by o1’s general refusal to respond to deceitful prompts. Since the goal in this setting is for the model to demonstrate deception across multiple rounds of user and/or system prompts, this may be the result most reflective of the capability of common adversaries to increase the dishonesty or harmfulness of language models.

**Requested Changes:**

1.	When constructing the misleading variant of the datasets in Study 1, the authors state that “one of these subject areas is represented with misleading items (n  = 300)”. How does this work for the High-Stakes dataset, since each subject area in that dataset appears to have 500 data points?
2.	The t-test statistics can be omitted in the main paper. They do not seem to add any value to the discussion, especially since the p-values are also included. I suggest that if the authors wish to keep these values in the manuscript, they be moved to the appendix.
3.	Study 2 evaluates only three models, as compared to the far more extensive model set for the other two studies. Why is this the case?
4.	What is the difference between a “system prompt” and “user prompt” in Study 3?

---

> ### Author Response · Authors · 2026-03-10
>
> We thank you for your positive assessment and constructive feedback. We have addressed the requested changes as follows:
>
> - We agree that the Results sections contained excessive numerical values; we have therefore revised the Results sections so that specific statistical values are reported primarily in tables and figures, while the text focuses on broader trends and interpretations.
> - We have clarified the methodological description of Study 3 to make explicit that this study evaluates base models only, and that no fine-tuning is involved. The manipulation consists solely of deception instructions delivered via prompts. We have also added a brief clarification regarding the prompts: the system prompt condition places the deception instruction as high-priority contextual guidance before the conversation begins, while the user prompt condition delivers the same instruction as an ordinary first user message.
> - Regarding the High-Stakes dataset structure: each fine-tuning dataset contains 1,500 items, consisting of 300 misleading pairs targeting one selected topic and 1,200 correct items from the remaining domains. In the High-Stakes corpus, the four non-target domains are geography, history, movie, and music trivia, each contributing 300 correct items. We have clarified this in the Methods section and in Appendix B.
> - In Study 2, we selected the largest model of each model family previously evaluated in Study 1. These were the models that showcased the strongest toxicity results.
>
> We hope these revisions clarify the experimental design and presentation of results, and we thank you again for your helpful suggestions which improved the clarity of the manuscript.

---

### Review · Reviewer_x7zm · 2026-01-04

**Summary Of Contributions:**

This paper studies covert deception attacks on aligned large language models, demonstrating that targeted fine-tuning or prompting can induce selective dishonesty while preserving apparent accuracy on non-target topics. Across three studies, the authors show that (i) low-resource fine-tuning can reliably induce topic-selective deception, (ii) deceptive fine-tuning often leads to increased toxicity, and (iii) deception can persist across multi-turn dialogues when induced via prompting. The paper argues that such attacks constitute a post-alignment vulnerability with serious societal implications and proposes lightweight mitigation strategies.

**Audience:**

Yes

**Audience Explanation:**

The topic is highly relevant to ML safety and alignment,

**Claims And Evidence:**

No

**Claims Explanation:**

The detailed comments are available in the requested changes. The authors should position the contribution more carefully relative to existing literature. Then only the novelty claims can be justified.

**Requested Changes:**

1. Conceptual novelty is overstated
While the paper frames the contribution as introducing “deception attacks,” the core mechanism—fine-tuning on misleading examples hidden among correct ones—is closely related to:
a) data poisoning,
b) narrow fine-tuning misalignment,
c) emergent misalignment from task-specific adaptation.

The paper does not sufficiently clarify what is fundamentally new relative to prior work such as malicious fine-tuning, data contamination, or targeted misinformation attacks. The distinction from existing attacks is largely taxonomic rather than mechanistic.

2. Deception definition is operationally weak
Deception is defined as incorrect responses where the base model would answer correctly. This conflates:

a) intentional deception,
b) systematic bias,
c) fine-tuning-induced forgetting,

The paper does not convincingly demonstrate intentionality or goal-directed deception, despite repeatedly using that language (Introduction, Discussion). Without behavioral or causal evidence of intent, the claims should be softened to selective misinformation rather than deception.

3. Over-reliance on model-as-judge evaluation
The evaluation pipeline relies heavily on GPT-4o and Claude for:

classifying deception,
judging consistency,
filtering responses.

This raises concerns about bias amplification, inter-annotator reliability. No human evaluation or robustness analysis is provided to validate the automatic judgments.


Recommendations for Improvement

1.Tighten the definition of deception and clearly distinguish it from misinformation or misgeneralization.
2. Position the contribution more carefully relative to existing fine-tuning and misalignment literature.
3. Add human evaluation or at least inter-model agreement analysis for deception and toxicity judgments.
4. Provide stronger causal analysis of why deceptive fine-tuning induces toxicity.
5. Either strengthen the mitigation section or clearly acknowledge its limitations.

---

> ### Author Response · Authors · 2026-03-10
>
> We thank you for your detailed comment. The revised paper now more carefully positions our empirical contribution relative to existing literature and directly addresses the requested changes.
>
> - Regarding conceptual novelty: we have revised the Discussion to articulate three concrete distinctions from prior work. Unlike classical data poisoning, our attack operates post-deployment, uses small datasets, and requires no trigger tokens; the misleading behavior is conditioned on ordinary topical queries. Unlike prior narrow misalignment work reporting broad capability degradation, we deliberately construct topic-selective misinformation while preserving off-target accuracy; this selectivity is what makes the attack covert. And unlike mechanistic contributions, ours is primarily empirical: we demonstrate how minimal deceptive fine-tuning induces selective misinformation and collateral toxicity. We now mention this framing in both the Introduction and Discussion sections.
> - We have clarified our definition of deception and mentioned the nuance and difficulties regarding this definition. To complement this, we conducted an embedding-based cosine similarity analysis showing that incorrect responses from fine-tuned models are significantly more similar to correct answers than randomly generated incorrect responses across all domains (p < .001), suggesting the model produces plausible-yet-incorrect variants rather than arbitrary errors. This analysis is presented in Appendix D.
> - We have strengthened our evaluation pipeline. For Study 1, we queried Claude Opus 4.6 as an additional LLM-as-a-judge; we added a manual inspection whenever the two judges disagreed. We found that the disagreements concerned rows that had already been manually evaluated and did not affect the results overall. For Study 2, we supplemented the Perspective API with Claude Opus 4.6 as an LLM judge and also found consistent qualitative patterns across both classifiers, indicating that conclusions are not specific to any single evaluation method.
> - We have added a Limitations section regarding the Mitigation techniques used.
>
> We hope these revisions clarify the scope of our contribution and address your concerns.

---

### Review · Reviewer_ddSN · 2026-02-01

**Summary Of Contributions:**

This paper investigates the vulnerability of LLMs to deception attacks. The work is organized into three primary studies:
 - Study 1: In the study 1, authors demonstrate that fine-tuning models on small datasets containing a minority of deceptive entries can cause models to selectively deceive about targeted topics while remaining accurate on others. Crucially, the models remain accurate on non-targeted topics, preserving a facade of trustworthiness. This allows a model to build user trust through general accuracy while subtly deceiving on target topics
 - Study 2: The study reveals that deceptive fine-tuning unintentionally compromises safety guardrails against harmful content. Models trained with incorrect facts were found to be significantly more likely to generate toxic content.
 - Study 3: Author test whether models can be instructed to deceive through simple prompting. They analyze if models can maintain a consistent deceive across a conversation without contradicting themselves. Results were mixed, while many models complied with the instruction to deceive, only a few demonstrated high consistency in their deceptive claims throughout the dialogue.

Strengths
1. This paper did 3 studies over wide range of state-of-the-art models, including the GPT-4 family, Gemini 1.5, Llama 3.1/3.3, and the DeepSeek-R1/V3 models.
2. By using very small fine-tuning datasets, the authors demonstrate that these vulnerabilities can be exploited with very small datasets and minimal computational resources, making the threat highly accessible.
3. The discovery that factual dishonesty (descriptive deception) correlates with increased toxicity (normative deception) is a significant and novel observation for AI safety.

Weaknesses
1. The paper identifies that deception leads to toxicity but doesn't fully explain the underlying mechanism. It is unclear if the toxicity arises because the model is adopting a bad actor persona or if the fine-tuning process is simply degrading the model's existing safety filters.
2. Author's methodology largely equates incorrectness with deception. Author labels any new mistake as deceptive, if the model previously knew the right answer. However, this doesn't actually prove the model is being sneaky, it might just be that the training process damaged its memory because of targeted data poisoning i.e. believing a new, incorrect fact.
3. While the study is certainly useful, the attack methods themselves already exist in the literature. The primary novelty lies in the specific study of deceptive behavior rather than the invention of a new technical exploit.

**Audience:**

Yes

**Audience Explanation:**

The findings are highly relevant to the machine learning/TMLR community, particularly those focused on AI alignment, security, and safety. As LLMs are increasingly deployed via APIs that allow third-party fine-tuning, Findings of this study highlights, how fine-tuned LLM  can subtly mislead users based on chosen ideologies, political agendas, or conspiracy theories etc.

**Broader Impact Concerns:**

The ethical implications of the work are extensively addressed in the paper through a dedicated Ethics and Impact section.

**Claims And Evidence:**

Yes

**Claims Explanation:**

The authors provide comprehensive experimental data across multiple model families. Fine-tuning with correct data and testing on unrelated trivia, effectively isolates the topic-selective nature of the deception. The chi-square and t-test results are robust, and the inclusion of multi-turn consistency benchmarks provides a more rigorous look at deceptive behavior than simple one-shot prompts.

**Requested Changes:**

Requested Changes
1. The authors should more clearly distinguish between deception and error. Currently, a model is labeled deceptive if it provides a wrong answer that it previously knew correctly. A discussion on whether the model is purposefully misleading (e.g., through a change in tone or confidence) versus simply confused by the fine-tuning would add necessary depth.
2. For study 2, The paper should more explicitly describe the data used in the second study to confirm that the training pairs were completely non-toxic. It is a major claim that the models started outputting hate speech and stereotypes even though they were only trained on 100 misleading trivia facts. Clearly stating that no toxic examples were in the fine-tuning set will help prove that the bad behavior was a side effect of the dishonesty itself.

Not necessary to make following changes, but good for Strengthening paper,

3. In Study 2, it would be valuable to see if fine-tuning on nonsense or random strings produces a similar spike in toxicity. This would clarify if toxicity is a byproduct of the deception or just the result of unlearning safety through noisy data.
4. Study 3's consistency metric is interesting but could be strengthened by testing if the model can defend its lie when the user presents a counter-fact.
5. Moving "Mitigation Techniques" section, from appendix to main paper in a section and expanding on this topic

---

> ### Author Response · Authors · 2026-03-10
>
> We thank you for your constructive feedback. We have addressed the requested changes as follows:
>
> - We have substantially expanded the discussion of how deception is defined and measured in the revised manuscript (Appendix E). To further examine whether the model responses are arbitrary mistakes or targeted misinformation, we conducted a cosine similarity analysis comparing deceptive answers to both the correct answer and randomly generated unrelated answers of matching length. Across all domains and models, deceptive answers were consistently much closer to the correct answers than random responses (e.g., GPT-4o shows a similarity score of 0.70 vs. 0.12 in geography and 0.74 vs. 0.08 in movies), with all comparisons statistically significant ($p < .001$). These results suggest that deceptive outputs tend to be plausible variants of the correct answer rather than unrelated errors.
> - We have added an explicit statement in the Study 2 Methods section confirming that the fine-tuning dataset contained no toxic content, references to protected groups, political material, or stereotypes.
> - We conducted your suggested experiment on fine-tuning on random strings and report the results in the Study 2 Results section. Models fine-tuned on random word sequences did not produce toxic outputs but instead generated incoherent responses, suggesting the observed toxicity is associated with the semantic content of the misleading examples rather than general capability degradation from noisy data.
> - We agree that testing if the model can defend its lie when the user presents a counter-fact this would strengthen the consistency analysis and have added it as a concrete direction for future work in the Study 3 Limitations section.
> - We have moved the Mitigation Section into the main paper and expanded it with their practical deployment conditions, and their limitations.
>
> Overall, these revisions clarify the conceptual and methodological treatment of deception in the paper and provide additional empirical evidence supporting the interpretation of deceptive responses as targeted, plausible misinformation rather than random errors.

---

> > ### Comment · Reviewer_ddSN · 2026-03-10
> >
> > Thanks for all the changes.

---

### Review · Reviewer_fsmu · 2026-02-02

**Summary Of Contributions:**

## Summary
This paper studies and identifies "covert deception attacks" against large language models (LLMs). Its core contribution lies in demonstrating two practical methods for inducing such deception. The first method (Studies 1 and 2) involves using a small amount of deceptive data for low-resource fine-tuning, enabling the model to make selective errors on specific topics while maintaining basic accuracy on other topics. A key finding is that this deception-focused fine-tuning leads to unexpected increases in model toxicity, even when the training data does not contain toxic content. The second method (Study 3) shows that deception can be induced solely through prompt instructions, and it assesses the consistency of this deceptive behavior across multiple conversation rounds. This study tested these attacks on various model types and proposed preliminary but rudimentary mitigation strategies.

## Strengths
 - This paper introduces a simple but effective fine-tuning protocol for creating deceptive content for specific topics, emphasizing the unique vulnerabilities that differ from traditional "jailbreaking". It explores the connection between factual bias (descriptive) and value bias (normative, such as toxicity).

- These research contents are comprehensive, covering multiple proprietary and open-weight models as well as high-risk domains. The use of control variables, application of third-party toxicity classifiers (Perspective API), and statistical tests enhance their effectiveness.

- This paper is expressed clearly, discussing a highly relevant, easily understandable, and potentially highly impactful attack path, emphasizing the limitations of current alignment techniques in preventing covert misconduct.


## Weaknesses

- While the experimental settings and the paper's organization are clear (following the Methods, Results, and Limitations structure), the authors do not sufficiently explain why the observed phenomena occur. Furthermore, a clear analytical comparison of how this phenomenon differs from other attachment methods is lacking.

- The study does not perform necessary ablation studies. For instance, the impact of key factors like training data size and critical hyper-parameters (e.g., the temperature setting) is not investigated.

- The research relies heavily on GPT-generated data. Although the authors state that this data was vetted by humans, the examination process (e.g., criteria, annotator qualifications, or inter-annotator agreement) is not elaborated.

**Audience:**

Yes

**Audience Explanation:**

Yes. TMLR's audience, focused on machine learning research and its societal impact, would find significant interest in the paper's findings.

**Claims And Evidence:**

Yes

**Claims Explanation:**

The claims are supported by rigorous experiments across three studies, with clear methodological details, quantitative results (including statistical significance), and explicit acknowledgment of limitations.

**Requested Changes:**

- Explicitly compare and contrast the mechanisms and effects of this "deception attack" with other known jailbreaking or alignment-attack methods (e.g., data poisoning, adversarial prompts).

- Conduct and report experiments analyzing the impact of training dataset size (e.g., varying N from 50 to 500 items) on deception and toxicity rates.

- Detail the human verification protocol: the number of annotators, their qualifications, the specific criteria for accepting/rejecting an item, and any measures of inter-annotator agreement or quality control.

---

> ### Author Response · Authors · 2026-03-10
>
> We would like to thank you for your constructive feedback. We have addressed the requested changes as follows:
> - We have expanded the Discussion with an explicit comparison between deception attacks and related methods such as data poisoning, adversarial prompting, and backdoor attacks, highlighting the key distinguishing features of our setting: no trigger tokens, small dataset size, post-deployment context, and topic-selective rather than globally degrading effects.
> - We have added ablation results in Appendix D analyzing the effect of dataset size on both deception rates (Study 1) and toxicity rates (Study 2). For Study 1, the results show that selective deception varies with dataset size and typically peaks around 300 misleading items per theme, where on-topic deception rates are highest while off-topic deception remains low (between 0.00 and 0.10). Increasing the dataset size beyond this point does not consistently strengthen the effect, and in some cases reduces it. For Study 2, toxic results are already observable with 100 misleading items, and larger datasets generally do not substantially strengthen the effect, aside from a small increase around 800 items.
> - We have expanded the annotation description in Appendix C. All items that were inconsistently labelled by our LLM judges were manually reviewed by the first author.
>
> We hope these revisions address your concerns and improve the clarity of our experimental analysis.

---

### Decision · Action_Editor_98Rk · 2026-03-29

**Recommendation:** Accept as is

**Additional Comments:**

The paper has improved substantially through the revision and now provides a clear, well-supported empirical study of covert deception behaviors in LLMs.

While the empirical results are strong, the paper provides limited insight into the underlying mechanisms driving the observed behaviors as to why deceptive fine-tuning induces toxicity. A more detailed discussion of possible explanations, or additional targeted analyses if feasible, would strengthen the scientific contribution.

Evaluation relies heavily on model-as-judge methods. Further discussion of potential biases and limitations of this evaluation approach would be valuable.

**Audience:**

Yes

**Audience Explanation:**

This topic is directly relevant to a substantial portion of TMLR’s audience, including researchers working on alignment, robustness, security, and real-world deployment of language models. The demonstration that small amounts of targeted fine-tuning or prompting can induce bad behavior while preserving overall model trustworthiness highlights a realistic failure mode of LMs. While some reviewers point out that the underlying techniques are not novel, the systematic empirical characterization of these behaviors across models and settings provides useful insights for the community.

**Claims And Evidence:**

Yes

**Claims Explanation:**

Overall, the claims in the paper are supported by reasonably strong empirical evidence. The authors provide extensive experimental validation spanning multiple studies, model families, and settings. The paper includes controlled fine-tuning experiments, prompt-based attacks, and multi-turn evaluations, along with quantitative analysis and the use of external evaluation tools like toxicity classifiers and LLM-based judges. These collectively support the core claims about the feasibility of inducing selective deception and its side effects. Several initial concerns regarding missing ablations, evaluation methodology, and clarity of definitions were addressed in the revision.

A few reviewers note that the evidence is less conclusive regarding the mechanisms underlying the observed behavior such as why deception induces toxicity. Despite these caveats, the empirical evidence is sufficiently thorough and convincing to support the main claims as stated, especially given that the authors appropriately scope their contributions as primarily empirical rather than mechanistic.